# Barefoot Footprint Detection Algorithm Based on YOLOv8-StarNet

**DOI:** 10.3390/s25154578

**Published:** 2025-07-24

**Authors:** Yujie Shen, Xuemei Jiang, Yabin Zhao, Wenxin Xie

**Affiliations:** 1College of Investigation, People’s Public Security University of China, Beijing 100000, China; ifsc_syj@163.com; 2Trace Examination Technology Department, Institute of Forensic Science of China, Beijing 100000, China; xuemei1103@sina.com; 3School of Automation and Software Engineering, Shanxi University, Taiyuan 030006, China; wenxin_xie@126.com

**Keywords:** barefoot footprint, criminal investigation, StarNet, YOLOv8, lightweighting, feature fusion, element-wise multiplication, low hardware requirements

## Abstract

This study proposes an optimized footprint recognition model based on an enhanced StarNet architecture for biometric identification in the security, medical, and criminal investigation fields. Conventional image recognition algorithms exhibit limitations in processing barefoot footprint images characterized by concentrated feature distributions and rich texture patterns. To address this, our framework integrates an improved StarNet into the backbone of YOLOv8 architecture. Leveraging the unique advantages of element-wise multiplication, the redesigned backbone efficiently maps inputs to a high-dimensional nonlinear feature space without increasing channel dimensions, achieving enhanced representational capacity with low computational latency. Subsequently, an Encoder layer facilitates feature interaction within the backbone through multi-scale feature fusion and attention mechanisms, effectively extracting rich semantic information while maintaining computational efficiency. In the feature fusion part, a feature modulation block processes multi-scale features by synergistically combining global and local information, thereby reducing redundant computations and decreasing both parameter count and computational complexity to achieve model lightweighting. Experimental evaluations on a proprietary barefoot footprint dataset demonstrate that the proposed model exhibits significant advantages in terms of parameter efficiency, recognition accuracy, and computational complexity. The number of parameters has been reduced by 0.73 million, further improving the model’s speed. Gflops has been reduced by 1.5, lowering the performance requirements for computational hardware during model deployment. Recognition accuracy has reached 99.5%, with further improvements in model precision. Future research will explore how to capture shoeprint images with complex backgrounds from shoes worn at crime scenes, aiming to further enhance the model’s recognition capabilities in more forensic scenarios.

## 1. Introduction

Barefoot footprint images in criminal investigations have become increasingly critical for case resolution, as they provide not only suspect characteristic information but also enhance investigators’ understanding of human physiological structures. This biological evidence strengthens investigative methodologies and enables precise suspect identification. As a pivotal biometric technology, footprint recognition complements other identification modalities to collectively deliver robust technical support for criminal case resolution. Footprints, serving as crucial physical evidence with high preservation rates at crime scenes, contain substantial information about foot skeletal structures, making them a highly promising biometric modality. Investigative analysis of barefoot footprints, when combined with complementary evidence, can provide substantial support for judicial proceedings. Moreover, barefoot image analysis demonstrates broad applicability in medical domains, particularly through pressure distribution assessments for diagnosing foot-related pathologies such as flat feet and high arches, offering precise clinical diagnostic references [1].

Early footprint studies primarily focused on the extraction of anatomical features of the foot, such as foot type classification (heel, arch, instep, and toe regions) and analysis of papillary ridge patterns. Anatomical feature analysis primarily relies on manual measurement, with accuracy influenced by the operator’s experience and requiring a significant amount of processing time. The papillary lines of barefoot footprints exhibit diversity, forming patterns such as arched, scoop-shaped, and bowl-shaped, accompanied by details such as starting points, bifurcations, and small hooks, providing a biological basis for individual identification. Such methods have played a crucial role in criminal investigations, such as estimating suspect height and analyzing behavioral patterns by combining geometric measurements of footprint length, palm width, and arch width with gait characteristics (stride length, stride width, and stride angle). However, this method requires high clarity of footprints; if the ridges are incomplete or blurred, identification becomes significantly more challenging. Additionally, gait characteristics are easily influenced by walking habits and ground material, resulting in poor reproducibility. These limitations have driven a paradigm shift in technology.

In recent years, deep learning-based features have supplanted traditional manual features in image recognition, achieving superior performance across multiple domains. Deep neural networks such as VGG [2], LeNet [3], and ResNet [4] have been widely adopted in image analysis, driving the transition of footprint recognition from conventional approaches to intelligent methodologies. These advancements have enabled breakthroughs in barefoot footprint identification by automating the extraction of hierarchical features without relying on manual feature engineering. For instance, convolutional neural network (CNN)-based algorithms [5] demonstrate significant advantages in barefoot footprint recognition. Continuous progress in deep learning has spurred the optimization of feature extraction and learning strategies to enhance recognition accuracy. A representative example is the ResNet50 architecture combined with Horizontal Pyramid Matching (HPM) [6] and a Separate Triplet Loss function [7], achieving a 96.2% top-1 recognition rate on large-scale footprint datasets while outperforming traditional methods across multiple evaluation metrics. This approach substantially improves both accuracy and efficiency in barefoot image recognition.

In computer vision, You Only Look Once (YOLO) revolutionized object detection by reframing it as an end-to-end regression problem that predicts target categories and locations in a single forward pass [8]. This innovation significantly improved detection speed and efficiency. Through iterative updates, the YOLO series has continually balanced accuracy and computational efficiency. The 2023 YOLOv8 version, developed by Ultralytics, achieves notable advancements in precision, inference speed, and usability. It incorporates an optimized network architecture with enhanced backbone and detection head designs, delivering faster inference speeds and improved accuracy under equivalent computational resources. These improvements enhance sensitivity to small object detection and performance in complex scenarios. By integrating advanced deep learning techniques and optimization strategies, YOLOv8 excels not only in traditional object detection but also extends its applicability to broader computer vision tasks. Its advantages have established it as a preferred solution in both industrial and academic domains, particularly for real-time applications requiring high computational efficiency. YOLOv8 abandons the traditional anchor-based design in favor of the anchor-free mechanism, which directly predicts the target center point and width and height. Considering the variable morphology of barefoot, anchor-free can avoid the limitation of preset frame shape and improve the localization accuracy of irregular contours, so YOLOv8 is considered for the barefoot detection task. The primary contributions of this paper are as follows:(1)The Star block in the StarNet [8] architecture is designed. An additional point convolutional layer is added behind the depth-separable convolution for extracting richer picture features in acquiring the picture feature information, and participates in the subsequent Star operation for element multiplication, which effectively improves the overall efficiency of the network.(2)The backbone feature interaction part of the Encoder layer of Transformer [9] is constructed. Richer semantic information is extracted through the AIFI module that incorporates multi-scale features and attention mechanisms, which improves the expressive ability of the model features, and maintains the computational efficiency while improving the detection accuracy. By further incorporating AIFI, the Gflops of the model was reduced by 0.4.(3)A new feature fusion method is proposed, using the feature modulation block FMB (Feature modulation block) in SMFANet [10] to deal with multi-scale features, integrating global and local information through different levels of features, reducing unnecessary redundant computations, and reducing the number of parameters and computational complexity. The Gflops of the model using this feature fusion method was reduced by 11.9 compared to the initial model.

This paper is organized as follows. In Section 2, the related work on barefoot image research is presented, including the research results of footprint detection techniques and the development status of the YOLO series. In Section 3, the optimization algorithm proposed in this paper is presented, which enables accurate prediction of barefoot footprint images. In Section 4, experimental results are shown and compared with previous methods. In Section 5, the main conclusions of this study are summarized and some future research directions are proposed.

## 2. Related Work

The history of footprint identification dates back to the 20th century. In early research, footprint identification was initially associated with forensic science and criminology and was primarily used as a means of evaluating evidence. In 1891, French anthropologist Alphonse Bertillon proposed a criminal identification method known as the Bertillon system, which included analyzing footprints at crime scenes. In 1977, British scientist C.W.L. Garner was the first to link image processing with footprint identification, studying how to use the texture and geometric features of images to distinguish footprints. Research during this period primarily relied on simple image processing techniques, such as edge detection [11] and shape analysis. In the 1980s, footprint identification began to integrate with pattern recognition technology, with researchers attempting to extract features from footprints using Fourier transforms and principal component analysis [12] (PCA).

Early research on footprint recognition abroad focused on traditional methods such as feature extraction and dynamic footprint and gait analysis. Feature extraction primarily involved analyzing characteristics such as foot length, foot width, and toe distribution. Researchers in the US analyzed footprint texture features (such as foot pressure distribution) and combined them with machine learning for classification. The purpose of gait footprint research is to achieve identification by analyzing changes in footprints (gait) during human walking. The important dataset Soton Gait Database [13] has driven research on gait and dynamic footprint features. In 2018, Japanese scholars proposed a multi-branch convolutional neural network [14] that segments footprint images into local regions and extracts features separately, thereby improving recognition accuracy. In 2025, scholars proposed the FootprintNet [15] network, which employs a footprint biometric recognition method using a Siamese Network to enhance the accuracy and efficiency of footprint recognition, distinguishing between similar and dissimilar footprint images. After conducting biometric data analysis on the performance of the footprint recognition system, its feasibility in practical applications can be validated, resulting in a compact and lightweight footprint recognition system. It is particularly suitable for operation on resource-constrained devices, such as portable scanners or resource-constrained forensic toolkits.

Early domestic research on footprint recognition primarily focused on traditional image-based approaches, such as geometric feature extraction and statistical modeling of footprints. Some Chinese scholars proposed a footprint recognition algorithm based on a dual-stream CNN [16], significantly improving the recognition rate of footprints in complex backgrounds. Domestic research on dynamic footprints has also been conducted, with scholars beginning to focus on the fusion analysis of dynamic footprints (gait), such as time series modeling of dynamic footprints using LSTM [17] (Long Short-Term Memory Network). Currently, research on dynamic footprints has gradually been applied in fields such as intelligent security and health monitoring. In 2021, Chinese scholars proposed a footprint image retrieval method based on an integrated deep neural network [18], using three models—VGG19, ResNet50, and DenseNet121—to extract footprint image features and establish a feature vector index database. This method demonstrates excellent performance in footprint image retrieval tasks, effectively assisting law enforcement personnel in quickly screening similar footprint images during case investigations and reducing the workload of manual retrieval. In 2023, scholars investigated multi-modal footprint recognition networks [19]. A lightweight multi-modal footprint recognition network based on feature fusion and attention mechanisms can more effectively extract key features from footprint images through feature fusion and attention mechanisms, thereby improving the accuracy and efficiency of footprint recognition while reducing parameters.

Research on footprint recognition began earlier abroad, but the methods were relatively simple and primitive. Although domestic research on footprint recognition started later, the research directions and content are relatively broad. The proposed research and methods all have their advantages and limitations, but they have laid the foundation for subsequent research. A specific summary is shown in Table 1.

YOLO is a target detection algorithm widely used in real-time target detection tasks due to its high speed and accuracy. Through continuous iteration and development, YOLO has been applied to numerous visual tasks, from the initial YOLOv1 [20] to YOLOv8, with its architecture and performance continuously improving. In previous versions, YOLOv9’s PGI (Programmable Gradient Information) mechanism increases training complexity, slows convergence speed, and lacks computational efficiency and processing speed. While YOLOv10’s advanced training strategies, such as self-supervised learning and zero-shot learning, offer theoretical advantages, they may increase training complexity and data preparation costs in practical applications. YOLOv11 has relatively higher training and deployment difficulties, requiring more data and computational resources to fully leverage its performance advantages. YOLOv12 introduces architectural changes such as the Region Attention Module (A2), Residual Efficient Layer Aggregation Network (R-ELAN), and FlashAttention, but the addition of these components increases computational overhead and slows down inference time. YOLOv8 has been optimized and improved in multiple aspects, not only inheriting the speed advantages of the YOLO series but also introducing a more efficient network architecture. By incorporating a sliding window mechanism similar to Swin Transformer [21] and the hierarchical structure of PVT [22] (Pyramid Vision Transformer), it can better extract local features from images, enhance global information modeling capabilities, and improve detection accuracy. Therefore, this paper selects a more suitable version of YOLOv8 for footprint detection.

Although YOLO has advantages in terms of speed, it still has some limitations. One of its main drawbacks is its relatively weak ability to handle fine-grained features and global information in images. Barefoot footprint images have relatively dense features, which are mainly concentrated within the footprint and contain a wealth of information. Since YOLO heavily relies on fixed-size grids for prediction, its performance in detecting dense objects is also suboptimal. Based on this insight, this paper proposes a feature fusion algorithm based on StarNet, incorporating an attention mechanism to improve YOLOv8’s performance in handling global information and dense features. This model is expected to enhance YOLO’s generalization capabilities in footprint recognition tasks and further promote the application of the YOLO series in more forensic scenarios.

## 3. Based on the YOLOv8-StarNet Feature Fusion Algorithm

YOLOv8 achieves a balance between accuracy and speed in barefoot detection tasks through innovative features such as lightweight architecture, anchor-free mechanisms, and multi-scale feature fusion. In terms of lightweight design, the C2f module in YOLOv8 reduces computational complexity by utilizing cross-stage local network structures while introducing richer gradient flow information. When detecting bare feet, the C2f module can more efficiently capture foot contours, texture, and other detailed features while improving inference speed. YOLOv8’s anchor-free detection method directly predicts the center of the target, simplifying model complexity and enhancing detection flexibility and accuracy, particularly suited for the diverse shapes and sizes of targets in barefoot images. The feature pyramid network in the Neck section of YOLOv8 enables more precise multi-scale detection. In barefoot detection, the model can simultaneously capture global foot contours (large-scale features) and local textures (small-scale features), improving the recognition rate of blurry or incomplete footprints by approximately 10%. The StarNet architecture enhances feature expression capabilities through simple element multiplication without increasing network width, helping to maintain model accuracy. StarNet uses multi-layer stacking to enable exponential growth in implicit dimensions, significantly improving nonlinear fitting capabilities and ensuring accuracy. In summary, combining YOLOv8 and StarNet enhances footprint detection performance, providing a reliable solution for barefoot recognition.

### 3.1. Overview Framework

Among the target detection algorithms, YOLOv8 is very fast for object detection and is able to detect targets at different scales, which is suitable for various complex scenes. However, the algorithm needs to use more network structures to improve the accuracy, which leads to a relatively large model size and increases the demand for computational resources and storage space. This study proposes an improved network architecture (Figure 1) based on YOLOv8, designed to reduce model parameters and computational complexity while enhancing detection accuracy. The backbone network is replaced with a shallower StarNet architecture featuring optimized Star Blocks. A pointwise convolutional layer is integrated into the feature extraction process to enable linear feature projection, followed by feature interaction through star operations, which facilitates learning of complex and abstract feature representations. To improve detection precision and speed, the SPPF [23] module is replaced with an Attention-based Interactive Feature Integration (AIFI) module that effectively fuses multi-scale features while capturing local-global contextual information. This modification enhances feature expressiveness while reducing computational redundancy. For feature enhancement and fusion, the Bottleneck structure in the C2f module is replaced with a Feature Modulation Block (FMB) comprising two components. Efficient Approximate Self-Attention (EASA), which captures non-local dependencies through a simplified self-attention mechanism to strengthen global information modeling. Local Detail Estimation (LDE), designed to extract fine-grained spatial details through dedicated operators, thereby improving the network’s capacity for detailed feature representation. The up-sampling in the architecture diagram enlarges the size of the feature map to twice that of the original one, and then the up-sampled feature map is spliced with the feature maps of different layers to realize feature fusion. The three Detect blocks in the figure act as the detection heads of the YOLOv8 model, which are connected to the back of the model to further process and predict the feature maps after Backbone and Neck processing, thus completing the whole target detection process and outputting the final detection results, including the categories and locations of the objects. By setting three Detect blocks of different scales, the model can detect objects of different sizes in the image at the same time, which improves the detection ability of the model for multi-scale objects, enhances the generalization and applicability of the model, and enables it to better complete the task of target detection in various scenarios. These architectural modifications collectively optimize the balance between computational efficiency and detection performance for barefoot footprint recognition tasks.

### 3.2. Optimized StarNet Architecture

StarNet is a lightweight neural network model based on the star operation (element-wise multiplication). By performing a simple element-wise multiplication operation, it can map inputs to a high-dimensional nonlinear feature space without increasing the network width (number of channels), thereby enhancing feature representation capabilities and helping to maintain model accuracy. This achieves efficient feature representation without increasing the network width, simplifying computations. When star operations are applied to neural networks and stacked across multiple layers, each layer introduces an exponential increase in implicit dimensional complexity. StarNet achieves exponential growth in implicit dimensions through multi-layer stacking. With just a few layers, star operations can approximate an “infinite-dimensional” state in a compact feature space, significantly enhancing nonlinear fitting capabilities while maintaining accuracy. This design enables StarNet to achieve richer and more expressive feature representations while maintaining computational efficiency. StarNet employs residual connections and attention mechanisms to enhance feature interaction and expressive power while maintaining computational efficiency. StarNet employs a four-stage hierarchical architecture, with the first step beginning with downsampling using a convolutional layer, which StarNet uses at each stage to reduce the resolution and double the number of channels. The second step is feature extraction of Star blocks which contain star operations in which the activation function is replaced with RELU6 and deep convolution is used at the end [24]. For efficiency, Layer Normalization is replaced with Batch Normalization after deep convolution to facilitate fusion operations later.

In the Star block, in order to better obtain the feature information of the picture, an additional point convolution layer is added behind the linear feature extraction of depth-separable convolution. Enhance the expressive power of the model by increasing the number of 1 × 1 convolution kernels. After stacking multiple point convolution layers, the depth of the model can be increased, and the depth and width of the model can be flexibly controlled, enabling the model to learn deeper feature representations. This operation can more fully capture the detailed information of all aspects of the picture and improve the feature expression ability after element multiplication, and the optimized structure is shown in Figure 2. Compared with existing network designs, StarNet does not have a more complex design, and hyper-parameter tuning and relies only on the high efficiency of star operation. Its design concept is significantly different from the traditional methods (such as convolution, the combination of linear layers and nonlinear activation) and mainly utilizes the implied high-dimensional features to improve the overall efficiency of the network. Because of its advantages of high efficiency, low latency, and strong expressive power, it is well suited for tasks such as image classification and target detection.

### 3.3. Attention-Based Intrascale Feature Interaction

The Rapid Spatial Pyramid Pooling (SPPF) module in the YOLOv8 backbone primarily serves to fuse multi-scale features and enhance feature map semantics for capturing contextual information across scales. While this design improves computational efficiency, it may compromise detection accuracy in certain scenarios. In contrast, the proposed AIFI module introduces a self-attention mechanism [25] to strengthen cross-scale feature interactions, prioritizing critical feature representations and thereby enhancing detection performance. Although structurally more complex, AIFI optimizes computational efficiency by eliminating redundant operations. Replacing SPPF with AIFI in the backbone reduces unnecessary computations and parameters while maintaining a balance between computational efficiency and model performance, as illustrated in Figure 3. By adding a 1 × 1 convolution layer before the AIFI module, the number of channels and feature fusion can be adjusted. After inputting the 2D feature map into this layer, a 1D feature vector is output. The basic internal structure of the AIFI module is shown in Figure 3.

The AIFI module is constructed based on the Transformer’s Encoder layer, which maps each token in the input sequence to its corresponding embedding vector containing rich semantic information. To incorporate positional awareness, positional encoding is added to explicitly represent the location of each token within the sequence. The core sub-layers consist of a multi-head self-attention mechanism and a feed-forward neural network [26]. The multi-head self-attention mechanism enables the model to capture interdependencies among all tokens in parallel, while the feed-forward network transforms the attention outputs into higher-level representations through computationally efficient parallel operations. Residual connections and layer normalization are applied to each sub-layer, allowing the Encoder to effectively encode both semantic and positional information of input sequences. This architecture enhances training efficiency and performance, particularly in processing long-sequence data.

### 3.4. Feature Modulation Block Based on SMFANet

The C2f module in YOLOv8 is a feature fusion module that further improves the performance and accuracy of the model by performing operations such as feature extraction and fusion. The Bottleneck in the module captures complex features in the image through a series of convolution, normalization, and activation operations to enhance the feature extraction capability of the model, but in some cases, it may not be able to adequately capture the details in the image, resulting in poor performance of the model in recognizing the target. To compensate for this limitation, we replace the feature modulation block FMB (Feature modulation block) in SMFANet with the Bottleneck structure in C2f.

The FMB consists of a Self-Modulated Feature Aggregation (SMFA) module and a Partial Convolutional Feedforward Network (PCFN), as shown in Figure 4. The Efficient Approximate Self-Attention (EASA) module in Self-Modulated Feature Aggregation (SMFA) is used for non-local feature interactions. In the computation process, EASA simplifies a large number of dot product and Softmax operations, which reduces the computational cost and improves the speed and efficiency of computation. Each person’s footprint characteristics are rich and unique. Therefore, to better capture the subtle feature details in barefoot images, LDE enhances the extracted local features and combines them with EASA through addition to better integrate the features. A partially convolutional feedforward network (PCFN) is used to improve the feature information generated by the SMFA layer. Merging these two components into a unified unit allows for the capture of more detailed information. This feature fusion module slightly increases the overall framework’s complexity but effectively improves the model’s accuracy while maintaining the YOLO series’ consistent lightweight characteristics.

In the SMFA module, a 1 × 1 convolution is first performed to extend the channel, then the channel is split into two parts as inputs to the EASA and LDE branches, and the features X and Y are processed in parallel through the EASA and LDE branches to produce the nonlocal feature X_t_ and the local feature Y_d_, respectively. Finally, the features are fused together by element-wise addition and fed into the 1 × 1 convolution in order to form a representative SMFA module Output. The low-frequency components are obtained by downsampling operations in EASA and fed into a 3 × 3 deep convolution to generate nonlocal structural information X_s_. Finally, we use modulation features to aggregate the input features X in order to extract the representative structural information X_l_. EASA prioritizes the exploration of nonlocal structural information, and LDE is used to capture the local features at the same time. The LDE uses a kernel size of 3 × 3 to extended depth convolution to encode the local information Y_h_ of the input feature Y. Two 1 × 1 convolutions and a hidden GELU activation are then used to generate the enhanced local feature Y_d_.

PCFN is used to further refine the SMFA-derived representative features. The PCFN interacts across channels by using a 1 × 1 convolution with a GELU activation function over an extended hidden space. The hidden feature F_p_ is then divided into two blocks {F_p1_, F_p2_}; F_p1_ is encoded with local contextual information using GELU through a 3 × 3 convolution and then connected to F_p2_ and fed into a 1 × 1 convolution for further feature blending and reduction in the hidden channel to the original input dimension.

## 4. Experimental Results and Analysis

The backbone of the StarNet architecture uses element multiplication to simplify computations. By stacking multiple layers, the implicit dimension grows exponentially, significantly enhancing nonlinear fitting capabilities. This approach maintains computational efficiency while yielding richer and more expressive feature representations. The AIFI module enhances the interaction capabilities of features at different scales through self-attention mechanisms, improving the model’s performance in detection and recognition tasks. FMB enhances image recognition capabilities by capturing subtle feature details and strengthening extracted local features. To validate the superiority of the proposed model in barefoot image recognition, this section will analyze its effective improvements in accuracy and lightweight performance through ablation experiments and comparative experiments.

### 4.1. Experimental Setup and Dataset

The experiments were conducted using the Windows 11 operating system with a 2.50 GHz i9-12900H CPU and an NVIDIA GeForce RTX 3060 GPU. A Python version of 3.8.19 was used for the training computations along with the PyTorch 2.0.0 deep learning framework and CUDA12.1.

The dataset used in the experiment is a self-constructed dataset, in order to complete the barefoot footprint recognition experiment, we use a professional footprint collection instrument to construct a barefoot footprint dataset. After we collected the initial footprints of 300 individuals, there were 8 barefoot images per person, totaling 2400 images after a simple data expansion by flipping horizontally and flipping vertically, which were captured as shown in Figure 5. The dataset is divided into training, testing, and validation sets according to the ratio of 6:2:2, with 60% of the data used for training, providing the model with enough data to learn features and patterns. This helps the model to better understand the distribution of the data, which improves its generalization ability. A remaining 20% each of validation and testing data provides enough data samples to accurately evaluate the model’s performance. This division ensures that the model can be adequately validated and tested during training to better tune hyperparameters and avoid overfitting.

Each image is set to a size of 640 × 640, and the training cycle for all models is set to 200 epochs. A resolution of 640 × 640 is highly versatile and strikes a good balance between accuracy and inference efficiency. This resolution avoids overloading the GPU while retaining sufficient image details for detection tasks, enabling faster inference speeds without compromising detection accuracy. The YOLOv8 architecture is optimized for the 640 × 640 resolution. This size can capture the detailed features of foot targets (such as contours, textures, and papillary lines) without causing a surge in computation due to excessive resolution. Typically, 200 epochs are sufficient for the model to learn and train adequately on the dataset, allowing the loss function to stabilize and the model’s performance metrics to reach a high level and stabilize. Considering the diverse morphology of bare feet, the SGD optimizer offers higher stability and better precision, making it more suitable for fine-tuning. Therefore, the SGD [27] optimizer was selected for the experiments, with specific training parameters shown in Table 2.

### 4.2. Evaluation of Indicators

In evaluating the performance of the model, we use the following widely adopted evaluation metrics: param (number of parameters), Gflops (billions of floating-point operations per second), mAP, P (precision), and R (recall). The precision rate P is the proportion of the predicted true class samples that are actually true by the model. Recall R is the proportion of all actual true class samples that the model successfully predicts to be true. mAP is the mean average precision, and mAP0.5 means that the IoU threshold is set to 0.5 when mAP is computed, i.e., a detection frame is correct if the IoU between the detection frame and the true target is greater than 0.5. Gflops is the number of floating-point operations performed by the model per second, and Params is the number of parameters in the model, which can be evaluated by the Params of the model, and Params evaluates the complexity and size of the model. The formulas for precision P, recall R, and mAP are given below:(1)$$P=\frac{TP}{TP+FP}$$(2)$$R=\frac{TP}{TP+FN}$$(3)$$\mathrm{mAP}=\frac{1}{c}\sum_{i=1}^{c}AP_{i}$$
where TP is the true case, FP is the false positive case, and FN is the false negative case.

### 4.3. Ablation Experiment

In order to evaluate and validate the effectiveness of the proposed improved algorithm, a total of four ablation experiments were conducted. Under the same environment and training parameters, and the results are recorded. In order to better compare the differences between the improvement points, the specific results of the ablation experiments after setting the same confidence level are shown in Table 3.

The first group is the original YOLOv8 algorithm; from the data in the table we can see that the YOLOv8 algorithm has a mAP value of 0.517 and a Gflops value of 29.3. The algorithm as a whole has a large number of parameters and a high model complexity. The original YOLOv8 backbone structure was replaced with the introduction of the improved StarNet structure in the experiment. This change makes the model more flexible and extensive in extracting features, which can fully obtain the detailed information of the image and improve the expression ability of the features, thus improving the detection accuracy. The experimental results show that as the mAP value rises to 0.726, there is a significant reduction in the number of Gflops and parameters, and the model becomes more lightweight. Through this experiment, we try to add the AIFI module of the Encoder layer in Transformer into the backbone of YOLOv8 as well, replacing the original SPPF module to reduce unnecessary calculations and parameters and keep the computational efficiency of the model. However, the experimental results show that in addition to the reduction in the number of Gflops and parameters, there is a significant decrease in the overall accuracy, with the mAP value decreasing to 0.534. This may be caused by the replacement of the spatial pyramid pooling, which leads to the model being less effective in feature fusion, and the model being less robust to changes in object scales, and unable to adequately capture the contextual information at different scales. To further validate the idea, the FMB is further used to replace the Bottleneck inside the feature fusion module C2f. By adding an attention mechanism and enhancing the feature extraction and interaction capabilities, the model is able to focus on richer information. This change helps the model to better integrate multi-scale feature information without adding more parameters and computation while maintaining the model performance. The experimental results show that the mAP value is further improved to 0.736, the Gflops is 17.4, and the number of parameters is kept near 6.5M without significant increase. Compared with the original YOLOv8 algorithm, our proposed model has a significant reduction in parameters and computation, the model lightweighting effect is significant, and the accuracy is substantially improved by 0.219. Through the above analysis of the results, we can see that the cumulative effect of these improvements is significant, and we can clearly see the contribution of each improvement point to the overall detection effect, which proves the validity of these improvement points.

To more clearly demonstrate the performance of the model proposed in this paper, we compared the test results of the initial model and the improved model by predicting barefoot images in the test set. We found that the improved model successfully detected some categories that the initial model failed to detect, with some of the results shown in Figure 6.

The original model is shown on the left and the improved model on the right in Figure 6. It can be seen that when the barefoot part of the image has unclear or incomplete content, the improved model can detect barefoot categories that the original model cannot detect. This shows that the improved model can better extract the features of the image. After integrating feature information at different levels, it enhances the model’s perception of the target shape and texture, ensuring the accuracy of recognition.

### 4.4. Comparative Experiments

In order to verify the effectiveness of the algorithms in this paper, the algorithms of the yolo series and other mainstream target detection models Faster RCNN [28], RT-DETR [29], and RetinaNet [30] are compared experimentally in this paper, and the experimental results are shown in Table 4. As can be seen from Table 4, the base model of YOLOv8 has a mAP value of 0.988, while the improved mAP value reaches 0.995, which improves the accuracy. The number of parameters is also reduced from 3.4 to 2.67, and the model becomes more lightweight. P (%) decreases and R (%) improves more obviously.

As shown in Table 4, the baseline YOLOv8 achieved a mAP of 0.988, while the improved model attained a higher mAP of 0.995 with reduced parameters (from 3.4 M to 2.67 M), demonstrating enhanced lightweight characteristics. Precision (P) exhibited a slight decline, whereas recall (R) improved significantly. Comparisons among YOLO-series variants revealed marginal accuracy differences, except for the older YOLOv5, which underperformed. YOLOv8 and YOLOv10 [31] achieved comparable accuracy, with YOLOv10 sacrificing marginal precision to reduce parameters and computational complexity for real-time efficiency. However, its performance may be constrained by reliance on large-scale annotated datasets, potentially limiting its applicability to our experimental setup. YOLOv12 [32], building upon YOLOv10’s lightweight design, further improved precision through an Area Attention (A2) module and FlashAttention [33] integration, optimizing attention computation and memory efficiency for global context modeling. Nevertheless, its dependency on specific GPU architectures increases hardware requirements and configuration complexity. Notably, YOLOv12’s A2 module and Residual Efficient Layer Aggregation Network (R-ELAN) prioritize generic object detection tasks, lacking robustness for small targets, orientation diversity, or complex backgrounds. Repeated pooling operations may lead to feature degradation, while the A2 module’s global receptive field struggles to capture fine-grained local details in footprint images. The model proposed in this paper does indeed have more parameters than YOLOv5 and YOLOv10, but YOLOv5 mainly reduces the number of parameters through lightweight designs such as separable convolutions and CSP structures, while YOLOv10 further compresses parameters through edge optimization techniques. Both are more inclined toward lightweight designs. Although the parameters of the model proposed in this paper are slightly higher than those of other versions, our model achieves a significant reduction in parameters and computational complexity compared to the original YOLOv8 algorithm, demonstrating notable lightweight optimization effects.

The algorithm in this paper is compared with other mainstream target detection algorithms, and the comparison results are shown in Table 5. From Table 5, it can be seen that the mAP of RetinaNet is 97.06%, which is slightly lower than other target detection algorithms, which may be due to the fact that although the feature maps in the lower layers of the algorithm contain detailed information, the small target features are lost after multiple downsampling. Since RetinaNet uses ResNet as the backbone network, multi-scale feature fusion and dense anchor frame prediction leads to a larger computational volume of the algorithm and higher training difficulty, which may not be able to satisfy the demand in scenarios with high real-time requirements. In addition, insufficient data and labels may make the model generalization ability appear degraded. Table 5 shows that the mAP of Faster R-CNN is 97.25%, with a slightly lower detection accuracy, which may be due to the fact that the feature maps extracted by this algorithm using convolutional networks are single-layered with a lower resolution, and the performance of the model may be limited when dealing with multiscale problems. As a two-phase detection algorithm, Fast R-CNN, with its larger number of parameters and more complex model, is slow in detection, meaning that its real-time performance is poor. From Table 5, we can see that RT-DETR has a higher recall of 0.99, which may be due to the fact that RT-DETR adopts a highly efficient hybrid Encoder, which is able to process multi-scale features quickly by decoupling the intra-scale interaction and cross-scale fusion, and the uncertainty-minimizing query selection strategy proposed by RT-DETR, which can provide a high-quality initial query for the decoder to ensure that the decoder is able to work from a more accurate initial state, which improves the recall of the model. However, due to the Transformer’s fully connected attention mechanism, RT-DETR needs to store large-scale key-value pairs, resulting in high memory consumption.

In order to better compare the performance between the model proposed in this paper and other models, we selected some models proposed in recent years to conduct some comparison tests, and the test results are shown in Figure 7.

In Figure 7, it can be seen that both YOLOv10 and YOLOv12 have incorrectly recognized boxes when detecting the same image. This is due to the fact that YOLOv10 removes the traditional NMS for prediction, which improves the efficiency but fails to effectively filter out the redundant predictions, which in turn leads to multiple frames for one object. YOLOv12 continues YOLO’s multi-scale prediction design, which detects different-sized objects through feature maps at different scales, but the feature extraction and fusion is not sufficient and fails to integrate the detection results of the feature maps at different scales effectively, resulting in redundant frame generation. Although RT-DETR successfully recognizes the category, the confidence level of the prediction frame is lower compared with the model proposed in this paper, and the model has a lower grasp of the judgment. The higher confidence level of the model proposed in this paper indicates that the model’s judgment of the predicted category is more accurate and the test results are more reliable.

The StarNet-based YOLOv8 optimization algorithm proposed in this paper improves the detection performance, and the improved mAP value reaches 0.995, with improved accuracy. The number of parameters is also reduced from 3.4 to 2.67, and the model becomes more lightweight. R (%) improves more significantly, and only P (%) decreases slightly. The number of parameters is also kept at a satisfactory level. Based on the above findings, for the barefoot footprint recognition task, the algorithm proposed in this paper outperforms other algorithms and is able to accomplish the detection task better.

## 5. Conclusions and Future Prospects

Aiming to address the problem of barefoot footprint recognition, this paper proposes an improved algorithm based on StarNet. This study proposes an enhanced StarNet-based algorithm for barefoot footprint recognition. The algorithm modifies the YOLOv8 framework by replacing its backbone with an optimized StarNet architecture, incorporating an additional pointwise convolutional layer to enhance feature extraction. Element-wise multiplication operations amplify local detail awareness, thereby improving feature expressiveness. The Rapid Spatial Pyramid Pooling module is replaced with an AIFI module that introduces self-attention mechanisms to strengthen multi-scale feature interactions. For feature fusion, FMB substitutes standard residual blocks, enabling non-local feature interactions and local detail extraction to boost detection performance. The proposed model achieves 99% mAP on the dataset, with Gflops reduced by approximately 12 compared to baseline YOLOv8, demonstrating effective lightweighting. Ablation and comparative experiments validate the model’s efficacy.

In this study, we design and implement a series of structural optimizations to improve the accuracy of the identification of people for barefoot footprints. Although our core work focuses on barefoot footprints, the proposed improvement strategies are promising and have cross-domain applications. These strategies show excellent results in recognizing footprints, a feature-concentrated and rich image, and their main core ideas and improvement directions are equally applicable to detecting other feature-concentrated images in a variety of scenarios. The improvement strategy in this paper has made some progress in terms of accuracy, but it is worth noting that the practical applications of the barefoot images studied in this paper are mostly in the field of criminal investigation. Generally, after there is a definite object of suspicion, the suspect is called for footprint acquisition and comparison, so the background of the image is mostly white and clearer. In future research, we can consider how to effectively obtain the complex background shoe print images in the crime scene to further promote the application of footprint recognition in more criminal investigation scenarios. In addition, although there are still more challenges regarding the existence of footprints, the more delicate detail feature extraction and model lightweighting the image has gives it obvious advantages in the application field of image recognition. On the one hand, it can increase the detection accuracy and efficiency of the model, thus improving the overall performance of the model; on the other hand, through the lightweight design of the model, it can reduce the computational cost, improve the application of the model in real-time detection, and solve the problem of its higher deployment cost in industrial scenarios, which is of certain research significance.

## Figures and Tables

**Figure 1 sensors-25-04578-f001:**
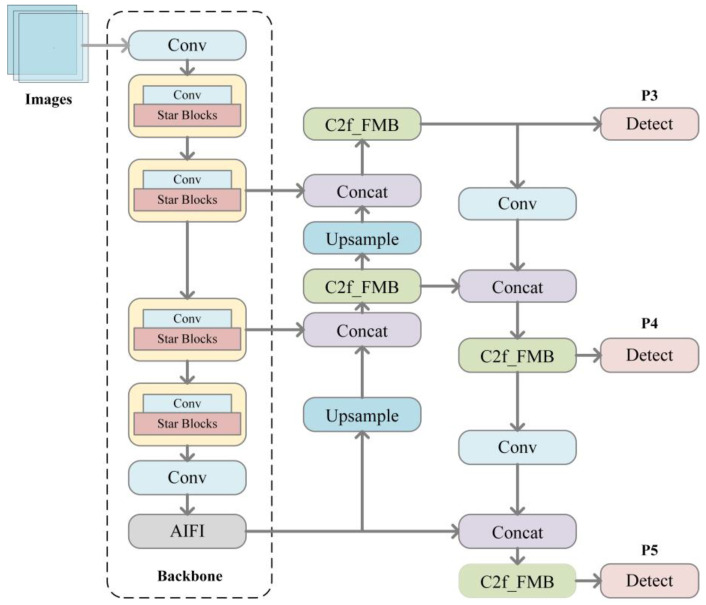
Overall network structure.

**Figure 2 sensors-25-04578-f002:**
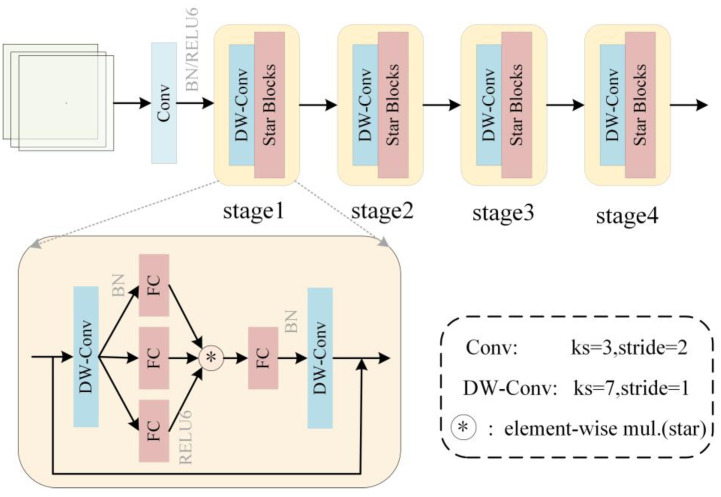
StarNet structure.

**Figure 3 sensors-25-04578-f003:**
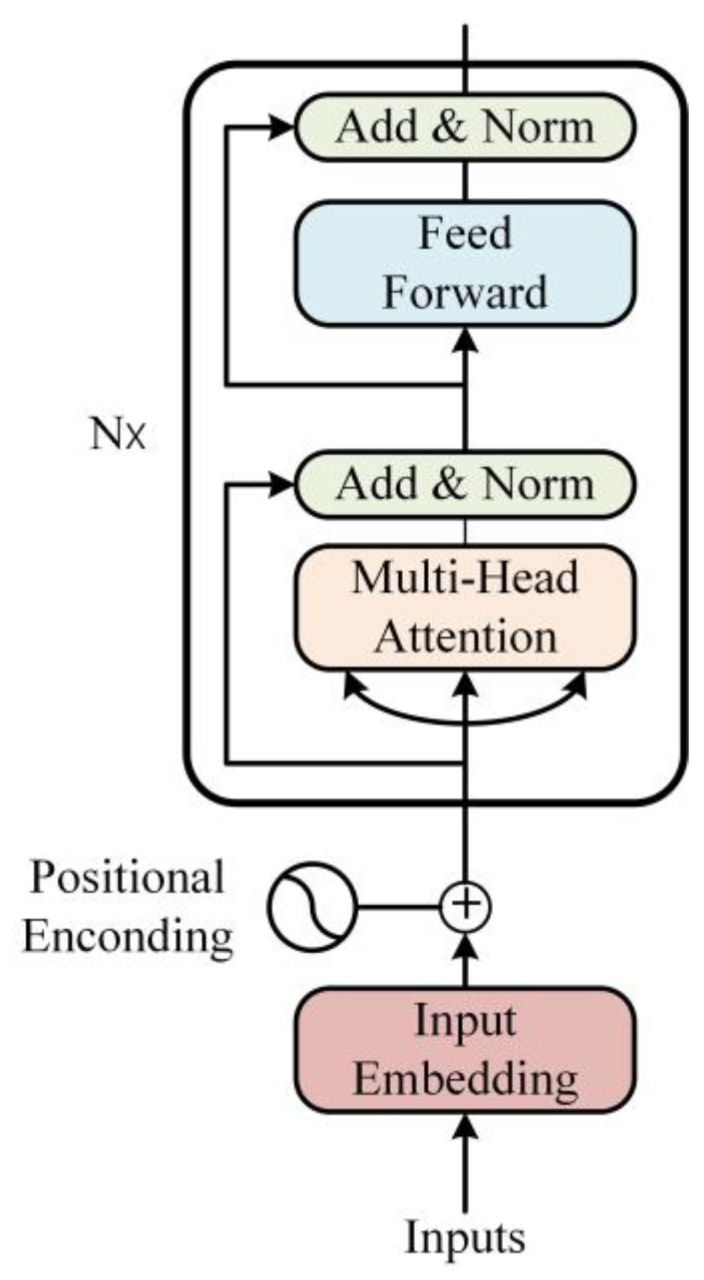
AIFI structure diagram.

**Figure 4 sensors-25-04578-f004:**
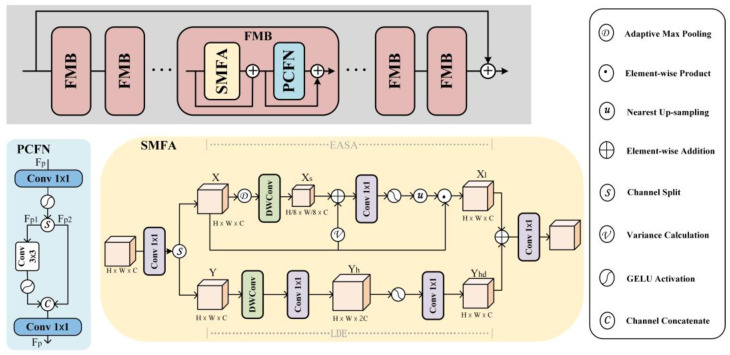
FMB structure.

**Figure 5 sensors-25-04578-f005:**
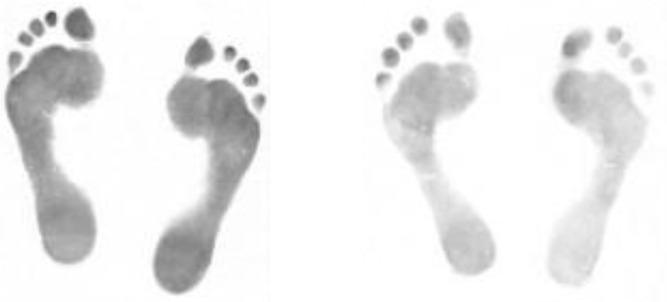
Barefoot pictures of some of the personnel.

**Figure 6 sensors-25-04578-f006:**
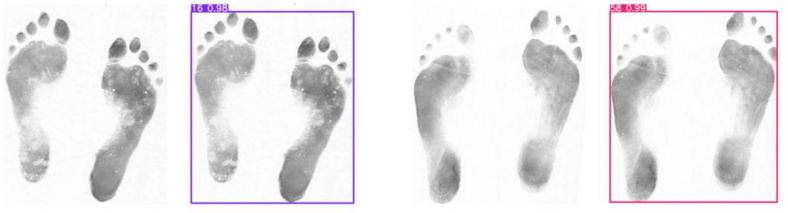
Test pictures.

**Figure 7 sensors-25-04578-f007:**
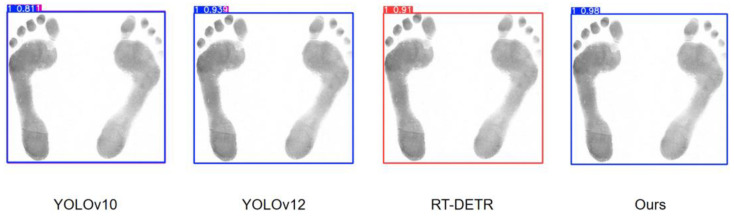
Test results of different models.

**Table 1 sensors-25-04578-t001:** Advantages and limitations of existing research methods.

Method Type	Research Results	Advantages	Limitations
Geometric feature extraction	Betty Yong measurement method (1891)	Intuitive and easy to explain; low computational complexity	Dependence on manual measurement
Texture/pressure analysis	CFPI Foot Pressure Image (2010)	Quantifiable pressure distribution	High sensor cost; poor ground adaptability
Dynamic gait recognition	Soton gait database; LSTM time series modeling	Strong anti-camouflage capabilities	Low accuracy at low resolutions
CNN static recognition	Dual-stream CNN (2015); FootprintNet twin network	High robustness in complex backgrounds	High-definition footprint images are required; large number of parameters

**Table 2 sensors-25-04578-t002:** Selected training parameters.

Parameters	Settings
image size	640
batch	8
epochs	200
optimizer	SGD
workers	4

**Table 3 sensors-25-04578-t003:** Improved point ablation experiments.

Method	StarNet	StarNet	FMB	P (%)	R (%)	mAP50	Gflops	Params (M)
Original	-	-	-	52.00	52.00	51.70	29.3	11.25
StarNet	√	-	-	73.00	73.00	72.60	18.3	6.75
StarNet + AIFI	√	√	-	53.70	53.70	53.40	17.9	6.66
Ours	√	√	√	**74.00**	**74.00**	**73.60**	**17.4**	**6.50**

**Table 4 sensors-25-04578-t004:** Comparison of detection performance of YOLO family of algorithms.

Types	YOLOv5n	YOLOv8n	YOLOv10n	YOLOv12n	OURS
P (%)	0.765	0.863	0.73	0.76	0.834
R (%)	0.91	0.877	0.913	0.899	**0.967**
mAP (%)	97.1	98.8	98.6	99.3	**99.5**
Gflops	5.4	9.9	7.2	7.0	8.4
Params (M)	2.16	3.4	2.31	2.68	2.67

**Table 5 sensors-25-04578-t005:** Comparison of detection performance of various types of target detection algorithms.

Types	RetinaNet	Faster RCNN	RT-DETR	OURS
mAP (%)	97.06	97.25	98.4	**99.5**
Gflops	239.02	377.067	104.8	**8.4**
Params (M)	42.53	142.81	32.60	**2.67**

## Data Availability

Data are contained within the article.

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
