# Peer review of "Barefoot Footprint Detection Algorithm Based on YOLOv8-StarNet"

_sensors, 2025, doi:10.3390/s25154578_

Round 1
Reviewer 1 Report
Comments and Suggestions for Authors
This article provides technical content to improve the performance of the YOLOv8 model to be suitable for practical use. However, some minor issues should be considered for revision. The details are as follows:
1. Please check the white spacing and capital letters at the beginning of sentences throughout the manuscript, especially Section 2. Related Work.
2. In-text citation should be included when referring to related content. For example, "In 2005, scholars Boulgouris et al. published in the journal Pattern Recognition..."
In addition, there is much research on deep learning, transfer learning, hybrid neural network architectures, and resource-constrained devices. It is suggested to add some works of literature in other fields, for example:
- Enhanced hybrid attention deep learning for avocado ripeness classification on resource constrained devices
3. Please provide more details about the dataset, such as how many images were collected per person (e.g., eight images per person). How were the training, testing, and validation datasets divided, and was there any balancing?
4. The author states that the dataset was divided into training, testing, and validation. They should be included if experimental results from the testing dataset are available. This will help to confirm the model's efficiency better. The testing dataset contains unseen data to help verify the model's detection/recognition capability in the real world.
Author Response
Reviewer: 1
Comments 1: Please check the white spacing and capital letters at the beginning of sentences throughout the manuscript, especially Section 2. Related Work.
Response 1: Thank you for the suggestions on the paper. We did find that the problem existed as you indicated. We have checked the entire beginning of the article to make sure that it all contains spaces and is all capitalized. In section 2 we have reworked all of the relevant work to ensure that it makes more sense in terms of content and formatting. The specific modifications are as described in Section 2. The details of the specific modification are marked in red.
Comments 2: In-text citation should be included when referring to related content. For example, "In 2005, scholars Boulgouris et al. published in the journal Pattern Recognition..."In addition, there is much research on deep learning, transfer learning, hybrid neural network architectures, and resource-constrained devices. It is suggested to add some works of literature in other fields, for example:
- Enhanced hybrid attention deep learning for avocado ripeness classification on resource constrained devices.
Response 2: Thank you for the suggestions on the paper. We did find that the problem existed as you indicated. Considering that the relevant content in Section 2 is less studied in the existing work, we have rewritten the content of Section 2 and added part of the literature on deep learning in order to better study the latest scientific research advances in recent years about this convenience. For example, the FootprintNet network proposed in 2025, which is a footprint biometrics approach through a twin network (Siamese Network), can improve the accuracy and efficiency of footprint recognition and distinguish between similar and dissimilar footprint images. After biometric data analysis of the performance of the footprint recognition system, its feasibility in practical applications can be verified and a compact and lightweight footprint recognition system is realized.2021 A footprint image retrieval method based on integrated deep neural networks is proposed. Footprint image features are extracted by using three models, VGG19, ResNet50 and DenseNet121, and a feature vector index database is built. The method has good performance in footprint image retrieval task, which can effectively help public security personnel to quickly screen similar footprint images in case investigation and reduce the workload of manual retrieval. A lightweight multimodal footprint recognition network about multimodal footprint recognition network based on feature fusion and attention mechanism is proposed. This network is able to extract key features in footprint images more effectively through feature fusion and attention mechanism, and can effectively improve the accuracy and efficiency of footprint recognition while reducing parameters.
Comments 3: Please provide more details about the dataset, such as how many images were collected per person (e.g., eight images per person). How were the training, testing, and validation datasets divided, and was there any balancing?
Response 3: Thank you for the suggestions on the paper. We did find that the problem existed as you indicated. We collected a total of 300 images from 300 people, eight images per person, for a total of 2400 images. The training, testing, and validation datasets are divided into a ratio of 6:2:2, which is a common division. 60% of the data is used for training, providing the model with enough data to learn features and patterns. This helps the model to better understand the distribution of the data, which improves its generalization ability. Validation and testing data, at 20% each, provides enough data samples to accurately assess the model's performance. This division ensures that the model can be adequately validated and tested during training to better tune the hyperparameters and avoid overfitting. The 6:2:2 division ratio is therefore a way to achieve a good balance between training, validation and testing, which can ensure that the model's performance is accurately assessed while fully utilizing the data resources.The specific modifications are as described in Section 4. The details of the specific modification are marked in red.
Comments 4: The author states that the dataset was divided into training, testing, and validation. They should be included if experimental results from the testing dataset are available. This will help to confirm the model's efficiency better. The testing dataset contains unseen data to help verify the model's detection/recognition capability in the real world.
Response 4: Thank you for the suggestions on the paper. We did find that the problem existed as you indicated. Therefore, In order to show the performance of the proposed model in this paper more clearly, we compared the test results of the initial model and the improved model by predicting the barefoot pictures in the test set, and some of the results are shown in Fig. 5. The original model is shown on the left and the improved model on the right. It can be seen that compared to the original model, the improved model has a higher accuracy in predicting the frames while ensuring the recognition accuracy, which indicates that the model is more deterministic about the recognition results and has a higher recognition accuracy. The specific modifications are as described in Section 4. The details of the specific modification are marked in red.
Reviewer 2 Report
Comments and Suggestions for Authors
The research proposed an integrated StarNet and YOLOv8 framework to enhance the footprint recognition. Although the framework has good research merit, the paper has various issues that need to be enhanced. Please refer to my comments below:
Comment 1. The paper’s title is not precise. The methodology is about the integration of YOLOv8 and StarNet and the utilization of element-wise multiplication and feature fusion. In addition, the application should be “barefoot footprint”.
Comment 2. Abstract:
(a) Clarify “state-of-the-art accuracy of 99.5%”. What are the accuracies of your work and the (ranges) existing works?
(b) Refer to “Future research will focus on enhancing the model’s robustness under diverse noise conditions to advance its practical deployment in forensic investigation scenarios.”. Does it mean the proposed work is limited to noiseless conditions?
Comment 3. Include more terms in the “Keywords” to better reflect the scope of the paper.
Comment 4. Section 1 Introduction:
(a) Discuss the performance of the manual barefoot footprint recognition approaches.
(b) Update “YOLO (You Only Look Once)” to “You Only Look Once (YOLO)”.
(c) Why was YOLOv8 considered? In the family of YOLO, older versions may work better in some applications.
(d) Is it a typo of “thesis” in “The primary contributions of this thesis are as follows”?
(e) Regarding the three research contributions, numeric results should be added to reveal the effectiveness of your work.
Comment 5. Once the acronyms are defined, do not redefine them.
Comment 6. Section 2 Related Work:
(a) Rewrite this section to discuss mainly recently published articles (2021-2025).
(b) Provide a concise summary of the methodologies, results, and limitations of the existing works. Add a table to facilitate the delivery of a summary.
Comment 7. Section 3 Based on StarNet feature fusion Algorithm:
(a) The heading of the section is not grammatically correct.
(b) Add an introductory paragraph before Subsection 3.1.
(c) Enhance the resolution of all figures. Zoom in on your file to confirm that no content is blurred.
(d) Figures 1 to 4, insufficient explanation is presented in the main paragraphs. Particularly, it is expected that the authors will justify the number of components, the sizes of components, and the linkages between components.
Comment 8. Section 4 Experimental Results and Analysis:
(a) Add an introductory paragraph before Subsection 4.1.
(b) A reference for the benchmark dataset is missing. In addition, provide a good summary of the dataset.
(c) Justify the ratio of the training, testing, and validation subsets.
(d) The role of Figure 5 is not clear. The caption should be enhanced to present subfigures (a) and (b).
(e) Justify the size of the images “640x640” and the number of epochs (200).
(f) Equation (1) is not “Accuracy” metric. Please correct the typo.
(g) Each “Method” in Table 2 is not clearly described. In addition, please extend your experiment with k-fold cross-validation and perform a statistical test to confirm that the proposed (integrated framework) achieved a better performance. In addition, share the time complexity of each model.
(h) For those existing works being compared with, ensure that in-text citations are provided.
Comment 9. Elaborate on the research implications and future research directions.
This paper contains many grammatical mistakes and improper use of English.
Author Response
Comments 1: The paper’s title is not precise. The methodology is about the integration of YOLOv8 and StarNet and the utilization of element-wise multiplication and feature fusion. In addition, the application should be “barefoot footprint”.
Response 1: Thank you for the suggestions on the paper. We did find that the problem existed as you indicated. We have reworked the title of the paper in line with the main research of the paper so that it can better match its content and meaning. We have revised the title section of the article. The details of the specific modification are marked in red.
Comments 2: (a) Clarify “state-of-the-art accuracy of 99.5%”. What are the accuracies of your work and the (ranges) existing works?
(b) Refer to “Future research will focus on enhancing the model’s robustness under diverse noise conditions to advance its practical deployment in forensic investigation scenarios.”. Does it mean the proposed work is limited to noiseless conditions?
Response 2: (a) Thank you for the suggestions on the paper. We include the value of accuracy in the abstract mainly to express the meaning that the model proposed in this paper has high accuracy, and does not state that this paper is the state-of-the-art model, maybe the content is not expressed very clearly. We have modified the relevant parts of the summary. The details of the specific modification are marked in red.
(b) Thank you for the suggestions on the paper. As this paper studies the practical application of barefoot images in criminal investigation more, generally in a determined object of suspicion, the suspect is summoned for footprint capture and comparison, so this paper is mainly studied in the absence of noise in the case of clearer barefoot images. With noise and other complex background images are mostly wearing shoes with shoe prints, due to the complex background of the shoe prints picture is difficult to clearly and effectively capture and compare, so this paper did not carry out in-depth research. We have modified the relevant parts of the summary. The details of the specific modification are marked in red.
Comments 3: Include more terms in the“Keywords”to better reflect the scope of the paper.
Response 3: Thank you for the suggestions on the paper. We did find that the problem existed as you indicated. We have included the more important term elemental multiplication in the keywords to better reflect the scope of the paper. We have modified the relevant parts of the summary. The details of the specific modification are marked in red.
Comments 4: (a) Discuss the performance of the manual barefoot footprint recognition approaches.
(b) Update “YOLO (You Only Look Once)” to “You Only Look Once (YOLO)”.
(c) Why was YOLOv8 considered? In the family of YOLO, older versions may work better in some applications.
(d) Is it a typo of “thesis”in“The primary contributions of this thesis are as follows”?
(e) Regarding the three research contributions, numeric results should be added to reveal the effectiveness of your work.
Response 4: (a) Thank you for the suggestions on the paper. We did find that the problem existed as you indicated. Therefore, we have expanded our discussion to include content related to manual barefoot footprint identification methods. Early manual footprint studies primarily focused on the extraction of anatomical features of the foot, such as foot shape classification (heel, arch, sole, and toe regions) and analysis of papillary ridge patterns. Among these, anatomical feature analysis primarily relied on manual measurement, with accuracy influenced by the operator's experience and requiring a significant amount of processing time. The papillary ridge patterns of barefoot footprints are highly diverse, forming patterns such as arched, scoop-shaped, and bowl-shaped, accompanied by details such as starting points, bifurcations, and small hooks, providing a biological basis for individual identification. The specific modifications are as described in Section 1. The details of the specific modification are marked in red.
(b) Thank you for the suggestions on the paper. We did find that the problem existed as you indicated. We have updated “YOLO (You Only Look Once)”to “You Only Look Once (YOLO)”. The specific modifications are as described in Section 1. The details of the specific modification are marked in red.
(c) Thank you for the suggestions on the paper. YOLOv8 abandons the traditional Anchor-Based design in favor of the Anchor-Free mechanism, which directly predicts the target center point and width and height. Considering the variable morphology of the barefoot (e.g., arch curvature, inter-toe gap), Anchor-Free avoids the restriction of the shape of the predefined box and improves the localization accuracy of irregular contours. While YOLOv5/v7 relies on predefined anchor frames, which is prone to frame matching bias when the foot posture is varied, so YOLOv8 is considered for the detection task of barefoot. The specific modifications are as described in Section 1. The details of the specific modification are marked in red.
(d) Thank you for the suggestions on the paper. We did find that the problem existed as you indicated. We have changed the terminology in the main contributions for better presentation. The specific modifications are as described in Section 1. The details of the specific modification are marked in red.
(e) Thank you for the suggestions on the paper. We did find that the problem existed as you indicated. Therefore, we have added relevant research results to the main contributions of this paper and supplemented some numerical results obtained from experiments to demonstrate the effectiveness of our work. The specific modifications are as described in Section 1. The details of the specific modification are marked in red.
Comments 5: Once the acronyms are defined, do not redefine them.
Response 5: Thank you for the suggestions on the paper. We did find that the problem existed as you indicated. We have checked the acronyms throughout the manuscript to ensure that there are no instances of multiple definitions of acronyms. The details of the specific modification are marked in red.
Comments 6: (a) Rewrite this section to discuss mainly recently published articles (2021-2025).
(b) Provide a concise summary of the methodologies, results, and limitations of the existing works. Add a table to facilitate the delivery of a summary.
Response 6: (a) Thank you for the suggestions on the paper. We did find that the problem existed as you indicated. We have rewritten Section 2 to make it more logical and informative, ensuring that it includes and discusses recently published articles (2021-2025). The specific modifications are as described in Section 2. The details of the specific modification are marked in red.
(b) Thank you for the suggestions on the paper. We rewrote the content of Section 2 and added a table summarizing existing research methods, results, and limitations to facilitate the presentation of the summary. The specific modifications are as described in Section 2. The details of the specific modification are marked in red.
Comments 7: (a) The heading of the section is not grammatically correct.
(b) Add an introductory paragraph before Subsection 3.1.
(c) Enhance the resolution of all figures. Zoom in on your file to confirm that no content is blurred.
(d) Figures 1 to 4, insufficient explanation is presented in the main paragraphs. Particularly, it is expected that the authors will justify the number of components, the sizes of components, and the linkages between components.
Response 7: (a) Thank you for the suggestions on the paper. We did find that the problem existed as you indicated. We have modified the content of the section heading to ensure that it is grammatically correct. The specific modifications are as described in Section 3. The details of the specific modification are marked in red.
(b) Thank you for the suggestions on the paper. We did find that the problem existed as you indicated. Therefore, we added a section introducing YOLOv8-StarNet before Section 3.1 to better demonstrate the unique advantages of this architecture and provide a reliable solution for footprint recognition. The specific modifications are as described in Section 3. The details of the specific modification are marked in red.
(c) Thank you for the suggestions on the paper. We did find that the problem existed as you indicated. We have checked the resolution of all the images in the text, adjusted them appropriately, and confirmed that there is no blurring of content. The details of the specific modification are marked in red.
(d) Thank you for the suggestions on the paper. We did find that the problem existed as you indicated. Therefore, we have made some simple changes in the paragraphs of Section 3. Some explanations have been added to the content in order to better illustrate some of the elements and linkages between the components. The details of the specific modification are marked in red.
Comments 8: (a) Add an introductory paragraph before Subsection 4.1.
(b) A reference for the benchmark dataset is missing. In addition, provide a good summary of the dataset.
(c) Justify the ratio of the training, testing, and validation subsets.
(d) The role of Figure 5 is not clear. The caption should be enhanced to present subfigures (a) and (b).
(e) Justify the size of the images “640x640” and the number of epochs (200).
(f) Equation (1) is not “Accuracy” metric. Please correct the typo.
(g) Each “Method” in Table 2 is not clearly described. In addition, please extend your experiment with k-fold cross-validation and perform a statistical test to confirm that the proposed (integrated framework) achieved a better performance. In addition, share the time complexity of each model.
(h) For those existing works being compared with, ensure that in-text citations are provided.
Response 8: (a) Thank you for the suggestions on the paper. We did find that the problem existed as you indicated. Therefore, we added a section before Section 4.1 to summarize the unique improvements of the YOLOv8-StarNet architecture. The experiments in Section 4 verify the effective improvements in accuracy and lightweight design of the model proposed in this paper for footprint recognition. The specific modifications are as described in Section 4. The details of the specific modification are marked in red.
(b) Thank you for the suggestions on the paper. The dataset used in this experiment is a self-built dataset, so there are no relevant citations. We have revised part of the overview of this dataset to more clearly illustrate the process of constructing the dataset. The specific modifications are as described in Section 4. The details of the specific modification are marked in red.
(c) Thank you for the suggestions on the paper. We divide the training, testing, and validation datasets into a ratio of 6:2:2, which is a common division. 60% of the data is used for training, providing the model with more than enough data to learn features and patterns. This helps the model to better understand the distribution of the data, which improves its generalization ability. Validation and testing data, at 20% each, provides enough data samples to accurately assess the model's performance. This division ensures that the model can be adequately validated and tested during training to better tune the hyperparameters and avoid overfitting. Thus the 6:2:2 division ratio is a way to achieve a good balance between training, validation and testing, fully utilizing the data resources while ensuring that the model performance is accurately evaluated. This division ratio is widely adopted in practical applications because it strikes a better balance between data utilization efficiency and model performance evaluation. The specific modifications are as described in Section 4. The details of the specific modification are marked in red.
(d) Thank you for the suggestions on the paper. We did find that the problem existed as you indicated. Figure 5 merely lists two random barefoot images of different individuals from the dataset. Upon closer inspection, it is evident that the two images are not identical and do not depict the same person. Therefore, there is no correlation between the two images; they may merely appear similar at first glance. To reduce any potential misinterpretation of the images, we have added additional explanatory notes to clarify their content. The specific modifications are as described in Section 4. The details of the specific modification are marked in red.
(e) Thank you for the suggestions on the paper. 640×640 is a highly versatile resolution that strikes a good balance between accuracy and inference efficiency. This resolution avoids overloading the GPU while retaining sufficient image detail for detection tasks, enabling faster inference speeds without compromising detection accuracy. The official pre-trained models for YOLOv8 (such as yolov8n.pt and yolov8s.pt) are all trained using the 640-pixel resolution. Using this size directly avoids feature mapping biases caused by size mismatches while leveraging pre-trained weights to accelerate model convergence. Additionally, YOLOv8's architecture is optimized for a 640x640 resolution. This resolution captures the detailed features of foot targets (such as contours, textures, and papillary lines) without causing a surge in computational load due to excessive resolution. YOLOv8 uses a dynamic loss function, whose effects require multiple iterations to fully unleash. Fewer epochs may cause the model to converge prematurely to a suboptimal solution. Typically, 200 epochs are sufficient for the model to undergo adequate learning and training on the dataset, stabilizing the loss function and enabling performance metrics (such as accuracy and recall) to reach high levels and stabilize. The specific modifications are as described in Section 4. The details of the specific modification are marked in red.
(f) Thank you for the suggestions on the paper. We did find that the problem existed as you indicated. We have corrected the incorrect indicator metrics in Equation (1) to ensure that they have the correct meaning. The specific modifications are as described in Section 4. The details of the specific modification are marked in red.
(g) Thank you for your suggestions for the paper. K-fold cross-validation is generally experimented with using raw data. Since the images in the dataset used in this paper have undergone some simple extensions, it is no longer a dataset containing only the original images. When we conduct this experiment, the initial model accuracy of the first group is 0.83, and the proposed model accuracy in this paper is 0.953; the initial model accuracy of the second group is 0.913, and the proposed model accuracy in this paper is 0.968; when we conduct the rest of the groups of experiments we find that it is difficult to keep the classification of the model balanced after the dataset has been expanded, the K-fold cross-validation leads to the bias of the evaluation results, and the results of some of the groups of the experiments have large fluctuations in accuracy from the results of the other groups of experiments. This may indicate that the dataset is not very suitable for this experiment. Therefore, to explain this, we added some notes to the dataset section. The specific modifications are described in Section 4. Details of specific modifications are marked in red.
(h) Thank you for the suggestions on the paper. We did find that the problem existed as you indicated. We have checked the models for which comparisons were made to ensure that they are correctly referenced in the text. The specific modifications are as described in Section 4. The details of the specific modification are marked in red.
Comments 9: Elaborate on the research implications and future research directions.
Response 9: Thank you for the suggestions on the paper. According to the reviewer's comments, We have revised some of the relevant parts of Section 6 to add some information about the significance of the research in this paper and future research directions in order to better clarify the work done in this paper. The details of the specific modification are marked in red.
Comments 10: This paper contains many grammatical mistakes and improper use of English.
Response 10: Thank you for the suggestions on the paper. We did find that the problem existed as you indicated. We have rechecked the entire article and corrected some grammatical errors and inappropriate use of English. The details of the specific modification are marked in red.
Reviewer 3 Report
Comments and Suggestions for Authors
This paper proposes the approach to footprint recognition based on enhanced StarNet model.
- StarNet reduces redundancy, but how do you maintain accuracy?
- How do you maintain accuracy from optimizing AIFI by eliminating redundant operations?
- What is the purpose of using feature fusion modlue? (section 3.4)
- If you use feature fusion module, isn't the computational complexity increasing?
- Some sentences miss the period mark.
- How do you construct footprint dataset?
- Is SGD the best optimizer?
- In Table 3, the number of parameters is bigger than other methods.
Author Response
Comments 1: StarNet reduces redundancy, but how do you maintain accuracy?
Response 1: Thank you for the suggestions on the paper. StarNet enhances the feature representation by mapping high-dimensional and non-linear input features with star operations, which helps to maintain the accuracy of the model. The residual linkage and attention mechanism are used to enhance the interaction and expression of features to improve the accuracy of the model, and StarNet can significantly improve the nonlinear fitting ability by stacking multiple layers to exponentially increase the implicit dimensionality and approach the "infinite dimensionality" state in a compact structure to ensure the accuracy. The specific modifications are as described in Section 3-2. The details of the specific modification are marked in red.
Comments 2: How do you maintain accuracy from optimizing AIFI by eliminating redundant operations?
Response 2: Thank you for the suggestions on the paper. We add a 1×1 convolutional layer at the input of the AIFI module to reduce the number of channels in the feature map and reduce the computational complexity.The 1×1 convolution can compress the dimension of the feature map and reduce the redundancy without losing important information. After that, the feature fusion module is used to better extract the features and ensure their accuracy. The specific modifications are as described in Section 3-3. The details of the specific modification are marked in red.
Comments 3: What is the purpose of using feature fusion modlue? (section 3.4)
Response 3: Thank you for the suggestions on the paper. The purpose of using a feature fusion model is mainly to capture as much feature information as possible. Each human footprint is unique and feature rich. Therefore, in order to better capture the subtle feature detail information in the barefoot images, the recognition accuracy is improved as much as possible by non-local feature interaction and local feature extraction. The specific modifications are as described in Section 3-4. The details of the specific modification are marked in red.
Comments 4: If you use feature fusion module, isn't the computational complexity increasing?
Response 4: Thank you for the suggestions on the paper. Generally feature fusion does increase the computational complexity, but we simplify a lot of dot product operations and softmax operations in the feature interaction part of the feature fusion module, which is used to reduce its computational cost and improve the computational speed and efficiency. The feature fusion module used in this paper only slightly increases the complexity of the overall framework, but effectively improves the accuracy of the model, which maintains the consistent lightweight characteristics of the yolo series. The specific modifications are as described in Section 3-4. The details of the specific modification are marked in red.
Comments 5: Some sentences miss the period mark.
Response 5: Thank you for the suggestions on the paper. We did find that the problem existed as you indicated. We checked the entire article and reworked some of the problematic sentences, adding some periods. The revisions were mainly focused on the third section. The specific modifications are as described in Section 3. The details of the specific modification are marked in red.
Comments 6: How do you construct footprint dataset?
Response 6: Thank you for the suggestions on the paper. Considering that barefoot pictures are difficult to collect and involve some privacy, we collected a total of 300 people's initial footprints after using specialized equipment, and each person had 8 barefoot pictures totaling 2,400 pictures after simple data expansion by horizontal and vertical flipping. This ensures that the model can be adequately trained to ensure the reliability of its performance. The specific modifications are as described in Section 4. The details of the specific modification are marked in red.
Comments 7: Is SGD the best optimizer?
Response 7: Thank you for the suggestions on the paper. Compared to other optimizers, SGD is more likely to converge to the global optimal solution. This is because the stochastic nature of SGD helps the model to jump out of the local optimal solution, especially in complex high-dimensional feature spaces. Considering the variable foot morphology of the barefoot, the SGD optimizer possesses higher stability and better accuracy, which makes it more suitable for fine tuning, and hence it is chosen to use the SGD optimizer. The specific modifications are as described in Section 4. The details of the specific modification are marked in red.
Comments 8: In Table 3, the number of parameters is bigger than other methods.
Response 8: Thank you for the suggestions on the paper. The model proposed in this paper does have an increase in parameters compared to yolov5 and yolov10, but the yolov5 model was proposed earlier, and is mainly designed to reduce the number of parameters through the lightweight design of deeply separable convolution and CSP structure. yolov10 further compresses the parameters through edge optimization techniques. Although the parameters of the model in this paper are slightly higher than the other versions, it reduces a large number of parameters compared with the initial model of yolov8, and still maintains the lightweight characteristics of the yolo series while improving the accuracy. The specific modifications are as described in Section 4-4. The details of the specific modification are marked in red.
Reviewer 4 Report
Comments and Suggestions for Authors
This manuscript proposed a network based on YOLOv8 for footprint detection. A Star block, incorporating element-wise multiplication, was integrated to the Conv blocks in the YOLO backbone. The SPPF module was replaced with an AIFI block that includes self-attention operations to extract richer features. The feature modulation blocks (FMB) were integrated into the C2f modules in both the FPN and PAN to enhance multi-scale feature fusion and extraction. The proposed method demonstrated improvements in both accuracy and inference speed for footprint detection compared to existing SOTA object detection models. However, the images used in the experiments were too simplistic, making them unsuitable for reliable performance evaluation. For a more rigorous evaluation, a dataset with footprint images containing more complex backgrounds is required.
- The term ‘Footprint recognition’ in the title is generally understood to mean identifying individuals or activities based on footprints. However, the proposed method focuses on footprint detections rather than recognition.
- Existing studies on feature fusion using the star operation (element-wise multiplication) should be analyzed.
- In Figure 2, it is unclear whether the modules inside the lower box represent the structure of Conv + Star Block or just the Star Block itself.
- Figure 3 is the structure of Transformer encoder, which seems to be incorrectly included. This module accepts 1D vectors as input, and cannot be directly applied to 2D feature maps.
- In Figure 5, the background appears to be a plain white color. A more complex background is necessary to effectively evaluate detection performance.
Author Response
Comments 1: The term ‘Footprint recognition’ in the title is generally understood to mean identifying individuals or activities based on footprints. However, the proposed method focuses on footprint detections rather than recognition.
Response 1: Thank you for the suggestions on the paper. We did find that the problem existed as you indicated. We have changed the identification in the title to detection, and given that the content of this paper deals primarily with barefoot images rather than images of footwear or other footprints, the title has been reworded to a more appropriate name in order to match its content. The details of the specific modification are marked in red.
Comments 2: Existing studies on feature fusion using the star operation (element-wise multiplication) should be analyzed.
Response 2: Thank you for the suggestions on the paper. Star operation is in fact only a very simple element multiplication, is not a very conducive to feature extraction and fusion method, so the relevant existing research is less, may mainly use the star operation or play a simplified operation effect, so did not mention too much relevant analysis. The details of the specific modification are marked in red.
Comments 3: In Figure 2, it is unclear whether the modules inside the lower box represent the structure of Conv + Star Block or just the Star Block itself.
Response 3: Thank you for the suggestions on the paper. We did find that the problem existed as you indicated. The overall structure module in the box represents the structure of Conv+Star Block, not only the Star Block itself, the dotted line in the diagram represents the whole Conv+Star Block structure. The dotted lines in the diagram represent the whole Conv+Star Block structure. Maybe because the box uses DW-Conv instead of Conv, it is easy to generate ambiguity, so the structure diagram is slightly modified to express its content and meaning more clearly. The specific modifications are as described in Figure 2. The details of the specific modification are marked in red.
Comments 4: Figure 3 is the structure of Transformer encoder, which seems to be incorrectly included. This module accepts 1D vectors as input, and cannot be directly applied to 2D feature maps.
Response 4: Thank you for the suggestions on the paper. We show the structure of the encoder simply to illustrate its internal structure and working principle, mentioning the relevant content, and does not involve the link with other parts of the operation. In the general framework of yolov8, We add a 1×1 convolutional layer in front of the AIFI module, which is used to adjust the number of channels and the fused features, and input the 2D feature maps into this layer to output the 1D feature vectors. Because the encoder is used in the overall framework, we describe the related contents here to facilitate readers to better recognize and understand. In order for the reader to better understand the framework, we have added some relevant notes. The specific modifications are as described in Section 3. The details of the specific modification are marked in red.
Comments 5: In Figure 5, the background appears to be a plain white color. A more complex background is necessary to effectively evaluate detection performance.
Response 5: Thank you for the suggestions on the paper. This paper examines the practical application of barefoot images in criminal investigation, generally in a determined object of suspicion, call the suspect for footprint collection and comparison. Taking into account the barefoot image collection is more difficult and involves some privacy issues, many complex background barefoot footprints are difficult to clearly and effectively capture and compare, so the use of clear white background image to provide more convenience for criminal investigation.The specific modifications are as described in Section 5. The details of the specific modification are marked in red.
Round 2
Reviewer 2 Report
Comments and Suggestions for Authors
The authors have made good enhancements to the quality of the paper; however, there are several important comments to be addressed:
Comment 1. Incomplete information in “Type of the Paper (Article, Review, Communication, etc.)”.
Comment 2. In the abstract:
(a) Incomplete numeric results in “parametric efficiency, recognition accuracy, and computational complexity”.
(b) The percentage of improvements by the proposed work compared with the existing work is missing.
Comment 3. Only one term is newly added to the keywords, which is insufficiently reflecting the scope of the paper.
Comment 4. Although the authors added justifications for the selection of YOLOv8, the explanation of not selecting other newer versions, i.e., YOLOv9, YOLOv10, YOLOv11, and YOLOv12, is missing.
Comment 5. In the revised article, excessive discussion was added to discuss content that was not up-to-date. Please ensure that the latest works focusing on the barefoot footprint detection algorithms are included.
Comment 6. The authors have not yet enhanced the resolution of all figures. Please zoom in on your file (e.g., 200%) to confirm that no content is blurred.
Comment 7. Figure 1 is not properly labelled and described. What are the parallel blocks in the upper left corner? What are the details of the two “upsample” blocks? What did the three “Detect” blocks refer to?
Comment 8. Subsection 3.2 is about “Optimized StarNet Architecture”, however, the authors did not fully explain how the architecture was optimally designed (along with the results of hyperparameter tuning).
Comment 9. The results of fine-tuning the architecture of the proposed model are missing.
Comment 10. A performance comparison between the proposed work and existing works (as well as baseline models) is missing.
Author Response
Comments 1: Incomplete information in“Type of the Paper (Article, Review, Communication, etc.)”.
Response 1: Thank you for the suggestions on the paper. We did find that the problem existed as you indicated. We have double-checked and reworded some of the information in the thesis type to make sure it is accurate. The details of the specific modification are marked in red.
Comments 2: (a) Incomplete numeric results in “parametric efficiency, recognition accuracy, and computational complexity”.
(b) The percentage of improvements by the proposed work compared with the existing work is missing.
Response 2: (a) Thank you for the suggestions on the paper. We did find that the problem existed as you indicated. We have scrutinized the relevant parts of the abstracts and revised the numerical results therein to be able to present their findings correctly. The details of the specific modification are marked in red.
(b) Thank you for the suggestions on the paper. We did find that the problem existed as you indicated. We have added a new percentage of improvement of existing work to better reflect the direction and research implications of this paper. We have modified the relevant parts of the summary. The details of the specific modification are marked in red.
Comments 3: Only one term is newly added to the keywords, which is insufficiently reflecting the scope of the paper.
Response 3: Thank you for the suggestions on the paper. We did find that the problem existed as you indicated. In order to better reflect the scope of this paper, we have added some new keywords to more accurately express their meanings. We have modified the relevant parts of the keywords. The details of the specific modification are marked in red.
Comments 4: Although the authors added justifications for the selection of YOLOv8, the explanation of not selecting other newer versions, i.e., YOLOv9, YOLOv10, YOLOv11, and YOLOv12, is missing.
Response 4: Thank you for the suggestions on the paper. We did find that the problem existed as you indicated. We have added some content notes about other versions to better explain why YOLOv8 was chosen for optimization in this paper. Among the existing versions, the PGI (Programmable Gradient Information) mechanism of YOLOv9 increases the training complexity, has slow convergence speed, and is deficient in computational efficiency and processing speed.YOLOv10's self-supervised learning, zero-sample learning, and other advanced training strategies, although theoretically more advantageous, may increase the training complexity and the cost of data preparation in practical applications.YOLOv11 is relatively more difficult to train and deploy, and requires more data and computational resources to fully utilize its performance advantages. yolOv12 introduces architectural changes such as the Area Attention Module (A2), Residual Efficient Layer Aggregation Network (R-ELAN), and FlashAttention, but the addition of these components leads to increased computational overhead and slower inference time. Therefore, in this paper, we choose the more suitable version of yolov8 for footprint detection. The specific modifications are as described in Section 2. The details of the specific modification are marked in red.
Comments 5: In the revised article, excessive discussion was added to discuss content that was not up-to-date. Please ensure that the latest works focusing on the barefoot footprint detection algorithms are included.
Response 5: Thank you for the suggestions on the paper. We did find that the problem existed as you indicated. We have reworked parts of the related work to reduce a bit of unnecessary discussion, focusing on the latest research directions regarding algorithms for barefoot footprint detection. The specific modifications are as described in Section 2. The details of the specific modification are marked in red.
Comments 6: The authors have not yet enhanced the resolution of all figures. Please zoom in on your file (e.g., 200%) to confirm that no content is blurred.
Response 6: Thank you for the suggestions on the paper. We did find that the problem existed as you indicated. We have rechecked all relevant chart content and increased the resolution to ensure that all graphs and symbols are clearly displayed when zoomed in.The specific modifications are as described in Section 3. The details of the specific modification are marked in red.
Comments 7: Figure 1 is not properly labelled and described. What are the parallel blocks in the upper left corner? What are the details of the two “upsample” blocks? What did the three “Detect” blocks refer to?
Response 7: Thank you for the suggestions on the paper. We did find that the problem existed as you indicated. We have reworked the content of Fig. 1 to better show its architecture.The parallel blocks in the upper left corner of the YOLOv8 architecture diagram represent the input images. The three Detect blocks on the right are Detect(P3), Detect(P4), and Detect(P5), which act as the detection heads of the YOLOv8 model and are connected to the back of the model for further processing and prediction of the feature maps after the Backbone and Neck processing to complete the whole target detection process. The final results are outputted, including the category and position of the object. By setting three Detect blocks of different scales, the model can detect objects of different sizes in the image at the same time, which improves the ability of the model to detect multi-scale objects, enhances the generalization and applicability of the model, and enables it to better complete the task of target detection in a variety of scenarios.The up-sampling operation of YOLOv8 adopts the nearest-neighbor interpolation algorithm, which can simply copy the pixel values to enlarge the size of the feature map to the original one, and the feature map can be upscaled to the original size. The size of the feature map is enlarged to twice the original size, and then the up-sampled feature map is spliced with the feature maps of different layers to realize feature fusion. This method is simple and fast in calculation and does not introduce new parameters, avoiding the increase in computational complexity and the risk of overfitting due to the increase in parameters. The specific modifications are as described in Section 3. The details of the specific modification are marked in red.
Comments 8: Subsection 3.2 is about “Optimized StarNet Architecture”, however, the authors did not fully explain how the architecture was optimally designed (along with the results of hyperparameter tuning).
Response 8: Thank you for the suggestions on the paper. We did find that the problem existed as you indicated. We have added a new section on optimizing the StarNet architecture so that readers can better understand the relevant changes made in this paper. We mainly enhance the representation of the model by increasing the number of 1×1 convolutional kernels. After stacking multiple point convolutional layers, the depth of the model can be increased, and the depth and width of the model can be flexibly controlled to enable the model to learn deeper feature representations. The specific modifications are as described in Section 3. The details of the specific modification are marked in red.
Comments 9: The results of fine-tuning the architecture of the proposed model are missing.
Response 9: Thank you for the suggestions on the paper. We did find that the problem existed as you indicated. We have newly added a test between the proposed model and the benchmark model in the ablation experiment section. In Fig. 6, the original model is shown on the left and the improved model on the right. It can be seen that when the barefoot portion of the image appears to have blurred content or incomplete footprints, the improved model is able to detect the barefoot category that the original model is unable to detect after the adjustment. This indicates that the improved model can better extract image features to capture more details. By integrating different levels of feature information, the model's ability to perceive the shape and texture of the target is improved, thus ensuring recognition accuracy. The specific modifications are as described in Section 4. The details of the specific modification are marked in red.
Comments 10: A performance comparison between the proposed work and existing works (as well as baseline models) is missing.
Response 10: Thank you for the suggestions on the paper. We did find that the problem existed as you indicated. We have added a new section on comparative experiments to compare the tests between the models proposed in this paper and the existing benchmark models. In Fig. 7, we can see that both YOLOv10 and YOLOv12 have incorrectly recognized frames when detecting the same image. This is due to the fact that YOLOv10 removes the traditional NMS for prediction, which improves efficiency but cannot effectively filter out redundant predictions, which in turn leads to the appearance of multiple frames for an object, and the model's own learning mechanism fails to fully ensure the uniqueness and consistency of the prediction in a given situation.YOLOv12 continues the design of YOLO's multiscale prediction, which detects objects of different scales by means of different scales of the feature maps. YOLOv12 continues the multi-scale prediction design of YOLO, which detects objects of different sizes through different scales of feature maps, but fails to effectively integrate the detection results of different scales of feature maps, resulting in the generation of redundant frames. RT-DETR successfully recognizes the category, but the confidence level of the prediction frames is lower compared with that of the model proposed in this paper, and the model has a lower grasp of the judgment. The higher confidence level of the model proposed in this paper indicates that the model's judgment of the predicted category is more accurate and the test results are more reliable. The specific modifications are as described in Section 4. The details of the specific modification are marked in red.
Reviewer 4 Report
Comments and Suggestions for Authors
The quality of the manuscript has improved through revisions. However, further improvements are still required.
- Although Figure 6 shows a slight increase in the score compared to the baseline model, this cannot be considered a significant performance improvement. It is recommended to present several cases where the proposed model successfully detects types that the baseline model failed to detect, and to analyze these cases.
- In Table 4, YOLOv5n shows the best performance in terms of GFLOPs and number of parameters, but the values corresponding to the proposed method are highlighted in bold. This may confuse readers.
- Line 557: "r(%)" -> "R(%)".
Author Response
Comments 1: Although Figure 6 shows a slight increase in the score compared to the baseline model, this cannot be considered a significant performance improvement. It is recommended to present several cases where the proposed model successfully detects types that the baseline model failed to detect, and to analyze these cases.
Response 1: Thank you for the suggestions on the paper. We did find that the problem existed as you indicated. In order to better demonstrate the superiority of the proposed model in this paper, we have added some relevant detection cases. In Fig. 6, the original model is shown on the left side and the improved model is shown on the right side. It can be seen that when the barefoot part of the image has blurred content or incomplete footprints, the improved model is able to detect the barefoot category that the original model is unable to detect. This shows that the improved model is able to better extract image features to capture more details. By integrating different levels of feature information, the model's ability to perceive the shape and texture of the target is improved, thus ensuring recognition accuracy. The specific modifications are as described in Section 4. The details of the specific modification are marked in red.
Comments 2: In Table 4, YOLOv5n shows the best performance in terms of GFLOPs and number of parameters, but the values corresponding to the proposed method are highlighted in bold. This may confuse readers.
Response 2: Thank you for the suggestions on the paper. We did find that the problem existed as you indicated. In order to enable readers to better understand the model presented in this paper, we have reworked the markup content of the bolded parts and bolded only the more prominent figures to better show the relevant content. The specific modifications are as described in Section 4. The details of the specific modification are marked in red.
Comments 3: Line 557: "r(%)" -> "R(%)".
Response 3: Thank you for the suggestions on the paper. We did find that the problem existed as you indicated. We have scrutinized the relevant content and made changes so that its content can be expressed more clearly. The specific modifications are as described in Section 4. The details of the specific modification are marked in red.